# High Prevalence of Hyperuricemia and Associated Factors among Zhuang Adults: A Cross-Sectional Study Based on the Ethnic Minority Population Cohort in the Southwestern China

**DOI:** 10.3390/ijerph192316040

**Published:** 2022-11-30

**Authors:** Lixian Zhong, Shun Liu, Xiaoqiang Qiu, Xiaoyun Zeng, Li Su, Dongping Huang, Xiaojing Guo, Jun Liang, Yu Yang, Xiaofen Tang, Yihong Xie

**Affiliations:** 1Department of Epidemiology and Health Statistics, School of Public Health, Guangxi Medical University, Nanning 530021, China; 2Department of Maternal, Child and Adolescent Health, School of Public Health, Guangxi Medical University, Nanning 530021, China; 3Key Laboratory of High-Incidence-Tumor Prevention and Treatment, Guangxi Medical University, Ministry of Education, Nanning 530021, China; 4Department of Sanitary Chemistry, School of Public Health, Guangxi Medical University, Nanning 530021, China

**Keywords:** Zhuang minority, hyperuricemia, gout, prevalence, risk factors, comorbidity

## Abstract

The highest prevalence of hyperuricemia was found in Zhuang minority adults in two national surveys in China, with only 1% Zhuang study subjects. However, the prevalence of hyperuricemia and the associated factors in Zhuang adults have not been well-addressed. A cross-sectional study was conducted to explore the prevalence of hyperuricemia and the common comorbidities, and the associated factors in Zhuang adults based on the Guangxi Ethnic Minority Population Cohort. Among 11,175 Zhuang adults aged 35–74 years, the age- and sex-standardized prevalence rate was 18.1% for hyperuricemia and 1.1% for gout. The standardized prevalence rate and awareness rate were 31.6% and 32.3%, respectively, for hypertension, and were 5.1% and 48.2%, respectively, for diabetes. High education level, history of coronary heart disease (CHD), hypertension, being a current drinker, high body mass index (BMI), central obesity, hyper-triglyceride (hyper-TG), hyper-total cholesterol (hyper-TC), hypo-high density lipoprotein cholesterol (hypo-HDL-C), and abnormal aspartate aminotransferase (AST) were risk factors, while smoking and diabetes were protective factors of hyperuricemia in males. Older age, being single/divorced, having a high education level, hypertension, drinking tea, high BMI, central obesity, hyper-TG, hyper-TC, hypo-HDL-C, and abnormal alanine aminotransferase (ALT) were risk factors in females. The high prevalence of hyperuricemia but low prevalence of gout and common comorbidities in Zhuang adults may be due to a lag effect of lifestyle changes. Health education and health management should be strengthened to prevent the progression of comorbidities, considering the lag effect and low awareness rate.

## 1. Introduction

Hyperuricemia, the most direct cause of gout, is a metabolic disease caused by a purine metabolism disorder that leads to continuously elevated levels of serum uric acid (SUA); it may occur when SUA is overproduced, or when there is a decrease in its excretion. Monosodium urate crystals may precipitate in the blood, kidneys, joint synovia, bursa, cartilage, etc., resulting in acute gout flares. The main clinical manifestations of gout are chronic gouty arthritis, which may accompany uric acid urolithiasis, tophi, and gout nephropathy. Previous studies have shown that hyperuricemia and gout were closely related to diabetes, hypertension, chronic kidney diseases, cardiovascular disease (CVD), and congestive heart failure [1]. Moreover, recent studies reported that gout was associated with chronic obstructive pulmonary disease (COPD), thromboembolism, atrial fibrillation, psoriasis [2], and cancer [3]. Hyperuricemia and gout have become a public health problem, due to their substantial effect on the health-related quality of life and severity of complications.

Previous studies showed that the prevalence of hyperuricemia had regional and racial variations due to the diversity of dietary patterns, socioeconomic conditions, and genetics [4,5,6,7], which in developed countries was higher than in developing countries [8]. Based on the diagnostic criteria of SUA > 420 μmol/L in males and SUA > 360 μmol/L in females, the prevalence of hyperuricemia in Austrian ≥ 50-year-old adults was 15.1% in males and 13.8% in females in 1985–2005 [9]. The prevalence in Japan was 25.8% (34.5% in males and 11.6% in females) in 2003 [10]. In the United States, the prevalence has hovered at around 20% in both males and females since 2007 [11]. In China, based on data from the China National Survey of Chronic Kidney Disease, the prevalence of hyperuricemia among ≥18-year-old adults was 8.4% (9.9% in males and 7.0% in females) in 2009–2010 [12]. With rapid economic development and changing lifestyles, the prevalence of hyperuricemia has increased rapidly in China. A systematic review reported that the pooled prevalence of hyperuricemia in China increased from 8.5% in 2001 to 18.4% in 2017. Guangxi Zhuang Autonomous Region (Guangxi) and Guangdong provinces are the areas with the highest prevalence of hyperuricemia in China [7]. A higher prevalence was shown in the Han cohort than in ethnic minorities [7]. Based on the Chinese Chronic and Risk Factors Surveillance (CCDRFS), the prevalence of hyperuricemia has increased to 11.1% (19.3% in males and 2.8% in females) in 2015–2016, and 14.0% (24.4% in males and 3.6% in females) in 2018–2019, using the new criteria of SUA > 420 μmol/L in both males and females; the highest prevalence (17.1%) was found in Zhuang minority adults [13]. However, limited studies focused on the prevalence of hyperuricemia in China, and most were based on hospital physical examination data. More than 90% of the study subjects were Han ethnicity, while only 1% were Zhuang minority in the CCDRFS. The prevalence of hyperuricemia in the Zhuang minority population has not been well addressed, and its associated factors have not been explored. 

Guangxi is located in the southwestern part of China. It is one of the five ethnic minority regions of China, and is home to more than 90% of the total Zhuang population in China. Using the Guangxi Ethnic Minorities Prospective Cohort of Chronic Diseases that was established in 2018–2019, we explored the prevalence, characteristics, and associated factors of hyperuricemia in Zhuang minority adults based on the cross-sectional baseline survey of the population cohort. Considering the potential high prevalence of hyperuricemia, the common comorbidities, e.g., gout, diabetes, coronary heart disease (CHD), hypertension, stroke, etc., were also explored.

## 2. Materials and Methods

### 2.1. Population

A cross-sectional study was conducted in 2018–2019 based on the Guangxi Ethnic Minorities Prospective Cohort of Chronic Diseases. This cohort recruited 14,561 adults aged between 35 and 74 years, with clustering sampling in 3 regions: Liangqing district of Nanning prefecture, Liunan district of Liuzhou prefecture, and Pingguo county of Baise prefecture. All of the adults in these three regions who met the following inclusion criteria were recruited: (1) ethnic minority adults aged between 35 and 74 years; (2) born in Guangxi; (3) living in Guangxi for more than 5 years before the cohort was established. As the cohort population in Liunan district and Pingguo county was recruited from the Liuzhou Steel Group and Pingguo Aluminum Corporation, respectively, the sera uric acid was untested. We only focused on the community-based data in Liangqing district. The procedure of study subject selection is shown in Figure 1. In total, 11,175 Zhuang minority adults were included. This study was approved by the Medical Ethics Committee of Guangxi Medical University (Number: 20170206-1). All participants voluntarily agreed to attend the study with written informed consent.

### 2.2. Measures

Face-to-face interviews using structured questionnaires and physical examinations were conducted by the trained investigators. The information collected included demographic characteristics (age, sex, occupation, marital status, and education level), health conditions (medical history, medication in the past six months, family disease history), and lifestyle (alcohol use in the past year, smoking in the past six months, and tea consumption in the past year). The items of physical examination included height, body weight, waist circumference, hip circumference, visceral fat grade, systolic blood pressure (SBP), and diastolic blood pressure (DBP). Blood pressure was measured twice for each participant, with mean values used in the analysis.

A fasting venous blood sample after an overnight fast of ≥10 h and a urine sample were also collected in the morning for routine testing and biochemical tests. The main items included triglyceride (TG), total cholesterol (TC), high density lipoprotein cholesterol (HDL-C), low density lipoprotein cholesterol (LDL-C), uric acid (UA), urinary protein, alanine aminotransferase (ALT), aspartate aminotransferase (AST), alkaline phosphatase (ALP), total bilirubin (TBil), direct bilirubin (DBil), and indirect bilirubin (IBil).

### 2.3. Relative Definitions

Hyperuricemia was defined as SUA > 420 μmol/L in males and females, according to the current Guidelines for the Diagnosis and Treatment of Hyperuricemia and Gout in China (2019) [14]. Alcohol drinking was defined as drinking alcohol at least once a week of more than 50 mL per occasion in the past year; this was further classified as moderate alcohol consumption (<25 g/d for males and <15 g/d for females) and excessive alcohol consumption (≥25 g/d for males and ≥15 g/d for females) [15]. Tea drinking was defined as drinking tea at least once a week of more than 150 mL per occasion in the past year. Smoking was defined as smoking at least 1 cigarette per day or ≥7 cigarettes per week in the past 6 months. Visceral fat obesity was defined as visceral obesity grade ≥ 10. Body mass index (BMI, kg/m^2^) was calculated using weight (kg)/height^2^, and was classified into four categories: underweight (BMI < 18.5), normal weight (BMI: 18.5–23.9), overweight (BMI: 24.0–27.9), and obesity (BMI ≥ 28) [16]. Central obesity was defined as a waist circumference > 90 cm in males and > 85 cm in females for Chinese adults according to the China Blue Paper on Obesity Prevention and Control [16]. Hyper-TG was defined as a TG level ≥ 2.3 mmol/L [17]. Hyper-TC was defined as a TC level ≥ 6.2 mmol/L [17]. Hyper-LDL-C was defined as an LDL-C level ≥ 4.1 mmol/L [17]. Hypo-HDL-C was defined as an HDL-C level ≤ 1.0 mmol/L [17]. Abnormal ALT was defined as an ALT level > 33 μ/L. Abnormal AST was defined as an AST level > 32 μ/L. Gout and CHD were defined as self-reported previous diagnoses by a doctor. Diabetes was defined as self-reported previous diagnosis by a doctor, or fasting blood glucose ≥ 7 mmol/L. Hypertension was defined as previously diagnosed as hypertension by a doctor, or SBP ≥ 140 mmHg, or DBP ≥ 90 mmHg.

### 2.4. Statistical Analysis

The data were analyzed using R version 4.0.3 (R Foundation for Statistical Computing, Vienna, Austria. http://www.r-project.org, accessed on 16 November 2020), with the “MASS” and “car” packages. As there are 12 ethnic minorities in Guangxi, and most of the cohort populations are Zhuang minority, this study only focused on the Zhuang ethnic minority group. Given that the prevalence and associated factors of hyperuricemia and gout are different in males and females, the analysis was conducted separately for each sex. The prevalence and proportion were calculated to describe the magnitude and the characteristics in the descriptive analysis. The prevalence was standardized by age and sex, based on the national census data in 2010 [18]. The proportion trend test was used to compare the prevalence by age group and education level. Univariable and multivariable logistic regression models were used to explore the associated factors of hyperuricemia, with odds ratios (OR) and 95% confidence intervals (CIs) provided. Variables with a *p*-value < 0.10 in the univariable analysis were included in the initial multivariable analysis. Multi-collinearity and interactions were checked between smoking and drinking alcohol. The Akaike information criterion (AIC) value was used to measure the goodness of model fit. All statistical analyses were performed as 2-tailed, with the level of significance set as 0.05.

## 3. Results

### 3.1. General Characteristics

A total of 11,175 Zhuang adults aged between 35 and 74 years were recruited. Their demographic characteristics are shown in Table 1. The median (interquartile range, IQR) age was 55 (47–64) years, and 45.3% were males. Females were less educated than males, with 69.9% achieving primary school or less, while it was 36.8% in males. For the health behavior in males, 41.5% were current smokers, and 21.2% reported excessive alcohol consumption (Table 2). In total, 32.7% of the study subjects were overweight, and 9.4% were obese.

### 3.2. Prevalence of Hyperuricemia and Gout, and Biochemistry Test Results

Among the whole cohort, 17.0% had hyperuricemia and 1.2% had gout. The age- and sex-standardized prevalence rates were 18.1% for hyperuricemia and 1.1% for gout. The age-specific prevalence in the 35–44, 45–54, 55–64, and 65–74 age groups were 17.6%, 17.2%, 15.8%, and 17.6%, respectively, for hyperuricemia, and 0.9%, 1.1%, 1.2%, and 1.6%, respectively, for gout. The prevalence rates of hyperuricemia and gout in males (29.5% and 1.7%, respectively) were higher than in females (6.6% and 0.8%, respectively).

The biochemistry test results are shown in Table 1. Except for HDL-C, the median levels of SUA, Glu, SBP, DBP, BMI, WC, TG, TC, LDL-C, ALT, and AST were all in ascending order by the normouricemia, hyperuricemia, and gout groups. The median level of SUA was 326.0 µmol/L (IQR: 269.0–391.0 µmol/L) among all study subjects; it was 372.0 µmol/L (IQR: 318.0–438.1 µmol/L) in males, and 291.0 µmol/L (IQR: 245.0–343.0 µmol/L) in females.

### 3.3. Common Comorbidities

The prevalence rates of self-reported previously diagnosed diabetes and hypertension were 3.3% and 15.3%, respectively. However, when considering fasting blood glucose and blood pressure test results, the prevalence rates of diabetes and hypertension increased to 6.2% and 37.7%, respectively. The age- and sex-standardized prevalence rates were 5.1% for diabetes and 31.6% for hypertension. Only 54.1% of individuals with diabetes and 51.3% with hypertension were aware of their condition. The standardized awareness rate was 48.2% for diabetes and 32.3% for hypertension. The other common comorbidities were CHD (1.1%), stroke (0.9%), COPD (0.5%), and cancer (0.3%). A part of study subjects had positive urine protein (10.2%), abnormal ALT (10.3%), and abnormal AST (8.2%) (Table 1). Among all of the study subjects, except for CHD which had no sex differences, all of the other comorbidities in males were more common than in females. Meanwhile, in the hyperuricemia group, the prevalence of diabetes, CHD, and hypertension in females were more common than in males (Figure 2).

### 3.4. Associated Factors of Hyperuricemia Stratified by Sex

The prevalences of hyperuricemia and associated factors among males are shown in Table 2. The prevalence of hyperuricemia was similar in the 35–44, 45–54, 55–64, and 65–74 age groups in males (χ^2^ = 1.78, *p*_trend_ = 0.182), while it increased with education level (χ^2^ =10.62, *p*_trend_ < 0.001). In the univariable analysis, high education level, history of CHD, hypertension, drinking alcohol, drinking tea, high BMI, central obesity, visceral fat obesity, hyper-TG, hyper-TC, hypo-HDL-C, abnormal ALT, and abnormal AST were all significantly associated with hyperuricemia, while smoking was protective. In the multivariable model, high education level, history of CHD, hypertension, drinking alcohol, high BMI, central obesity, hyper-TG, hyper-TC, hypo-HDL-C, and abnormal AST remained significant, while smoking and diabetes were protective factors.

Among females, the prevalence of hyperuricemia increased with age (χ^2^ = 39.52, *p*_trend_ < 0.001) and education level (χ^2^ = 4.36, *p*_trend_ = 0.037). Older age, being single or divorced, having a high education level, history of diabetes, CHD, hypertension, moderate alcohol consumption, drinking tea, high BMI, central obesity, visceral fat obesity, hyper-TG, hyper-TC, hyper-LDL-C, hypo-HDL-C, positive urine protein result, abnormal ALT, and abnormal AST were all significantly associated with hyperuricemia in the univariable analysis. In the multivariable model, older age, single/divorced, high education level, hypertension, moderate alcohol consumption, drinking tea, high BMI, central obesity, hyper-TG, hyper-TC, hypo-HDL-C, and abnormal ALT remained significant, while diabetes, history of CHD, visceral fat obesity, hyper-LDL-C, positive urine protein result, and abnormal AST did not (Table 3).

## 4. Discussion

This population-based study showed that the age- and sex-standardized prevalence rates were 18.1% for hyperuricemia and 1.1% for gout in the 35–74 age group of Zhuang minority adults. The prevalence of hyperuricemia was similar in the 35–44, 45–54, 55–64, and 65–74 age groups; however, the prevalence of gout increased with age. The prevalence of hyperuricemia was higher than the general level in China found from two national surveys [13], and higher than Japan in 2016–2017 (13.4%) [19]. Meanwhile, the prevalence of gout was relatively low compared to that of other studies, such as the USA (3.9%) [11], UK (2.5%) [20], and China (1–3%) [21]. The high prevalence of hyperuricemia but low prevalence of gout was inconsistent with the previous findings from a systematic review, which reported that the prevalence of hyperuricemia and gout in Guangzhou in 2008 and Nanjing in 2014 were both higher than the general level in China [21,22,23]. Consistent with previous studies [21], the prevalence rates of hyperuricemia and gout in males were both higher than in females.

Accumulating evidence shows that hyperuricemia is a prerequisite of gout [24], and is associated with increased risks of diabetes, hypertension, stroke, CHD, COPD, etc. [1,2,3,25,26,27]. In this study, the age- and sex-standardized prevalence rates were 31.6% for hypertension and 5.1% for diabetes, which were both lower than those for Chinese adults (37.2% for hypertension and 12.4% for diabetes) [28,29]. The prevalence rates of CHD, COPD, stroke, and cancer were also lower than those for China [30,31]. Hyperuricemia is an earlier-onset metabolic disorder than gout, hypertension, hypertriglyceridemia, and diabetes mellitus [32]. The relatively high prevalence of hyperuricemia but low prevalences of gout, hypertension, diabetes, and other common comorbidities may be due to the earlier effects of lifestyle changes on serum uric acid. Moreover, the different prevalences may be due to the different characteristics of the study subjects. Guangxi is located in the southwestern part of China, and has a lower economic level and poor health literacy. Only 9.4% of our study subjects were obese, and most were rural residents; meanwhile, more than 15% were obese, and over one-third were urban residents in the two national surveys. However, over 21% of our study subjects had excessive alcohol consumption, which is a risk factor for hyperuricemia [33]. In addition, the diagnosis of diabetes in our study was only based on previous diagnoses and fasting blood glucose test results, while the study in China also considered the 2-h plasma glucose level and HbA1c [29]. Thus, a part of potential diabetes cases may not have been detected in our study.

It should be noted that the standardized awareness rate of hypertension in our study was also lower than that for China (32.3% vs. 36.0%) [28], while for diabetes it was higher than that for China (48.2% vs. 36.7%) [29]. A study in Italy also reported that the awareness rate for hypertension was lower than that for diabetes [34]. The relatively high awareness rate for diabetes may due to the smaller denominator as part of the potential diabetes cases that were not detected in our study. However, the awareness rate of hypertension was low; as our study subjects were all minority ethnicities with lower education who mainly lived in rural areas, these were factors associated with the lower awareness rate [28]. Further evidence could be shown in the prevalence of hypertension in the hyperuricemia group being much higher than that for China (48.1% vs. 43.6%) [12]. As elevated SUA levels can increase the risk of hypertension [35], a low awareness rate is unfavorable for preventing the progression of comorbidities. Especially in females, who had a lower education level compared to males, the common comorbidities of diabetes, CHD, and hypertension in the hyperuricemia group were more common than in males.

Consistent with previous studies [9,36], the prevalence of hyperuricemia had an increasing trend by age in females, while no such trend was evident in males. Females were found to have a lower risk of hyperuricemia than males before the age of 50, while the prevalence increased significantly after menopause [37]. Similarly to the previous findings [38], high education level was a risk factor for hyperuricemia, while smoking was a protective factor [39]. Higher education individuals are more often engaged in mental work, with relatively less physical activity, more social activities, and a higher purine diet as well as drinking. As expected, hypertension, drinking alcohol, high BMI, central obesity, hyper-TG, hyper-TC, and hypo-HDL-C were all risk factors for hyperuricemia in both males and females, which was consistent with the studies of other regions in China, and with other ethnic groups [40,41]. The associations between hyper-LDL-C, diabetes, and hyperuricemia were inconclusive. Some studies showed that hyper-LDL-C was a risk factor for hyperuricemia [12,36], while no association was found in other studies [40,41], nor in our study. One study in China found that diabetes was a risk factor for hyperuricemia in females but not in males [42], while no such association was found in another study [40]. We found that diabetes was a protective factor for hyperuricemia in males, but found no such association in females. It was reported that SUA levels in diabetes cases were lower than those in non-cases [43]; diabetes itself may reduce the risk of hyperuricemia [24]. However, diabetes may increase the risk of hyperuricemia through accompanying comorbidities, as it is usually accompanied with obesity, hypertension [44], and decreased renal function [45], which are all risk factors for hyperuricemia.

## 5. Limitations

Although this study was conducted on the basis of a population cohort with a large sample size, and we measured the prevalence of hyperuricemia, hypertension, and diabetes by physical examination and biochemistry test results, there are still some limitations that should be mentioned. Firstly, the study areas of the cohort study were selected by convenience sampling. Moreover, the data used in this analysis came from a cross-sectional survey of baseline data. The order of contracting hyperuricemia and comorbidities, and their associated factors, could not be determined. Secondly, only one blood sample was used in the diagnosis of hyperuricemia, due to the high cost and difficulty of collecting blood samples. Thus, the prevalence of hyperuricemia in this study may be overestimated. Finally, the cohort was established in three districts but, due to the sera uric acid, were not tested in Liunan district and Pingguo county. We only focused on the cohort in Liangqing district, where most study subjects were rural residents. This may not represent the entire Zhuang population in Guangxi. Generalizations of the study results should be made with caution.

## 6. Conclusions

The prevalence of hyperuricemia in Zhuang minority adults in Guangxi was relatively high, while the prevalence of gout, hypertension, diabetes, and other common comorbidities was lower than the average level of China. As hyperuricemia plays an upstream role in the development of cardiovascular disease, the high prevalence of hyperuricemia but low prevalence of common comorbidities may be due to the early effects of lifestyle changes on serum uric acid. The lower awareness rate of hypertension implicated poor health literacy, and the risk of development to common comorbidities was high. Health education and health management should be strengthened to prevent the further progression of comorbidities.

## Figures and Tables

**Figure 1 ijerph-19-16040-f001:**
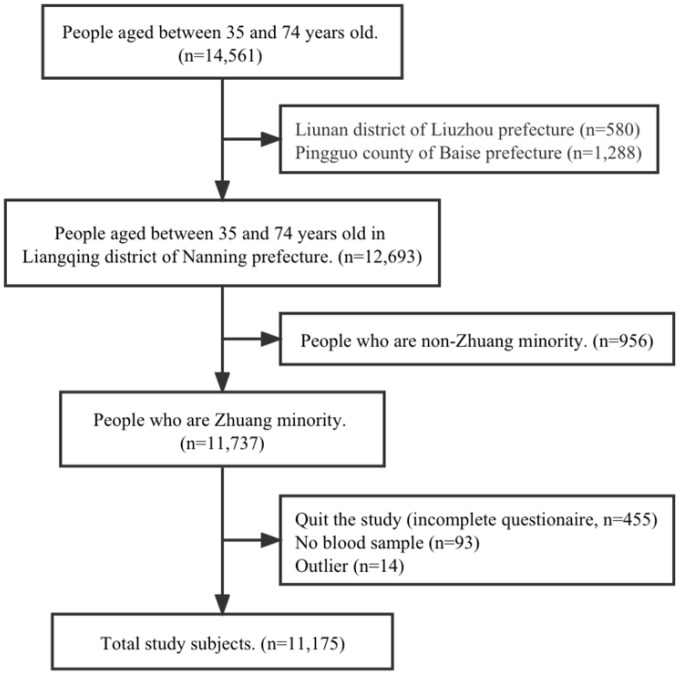
Flow chart for the study subject selection process.

**Figure 2 ijerph-19-16040-f002:**
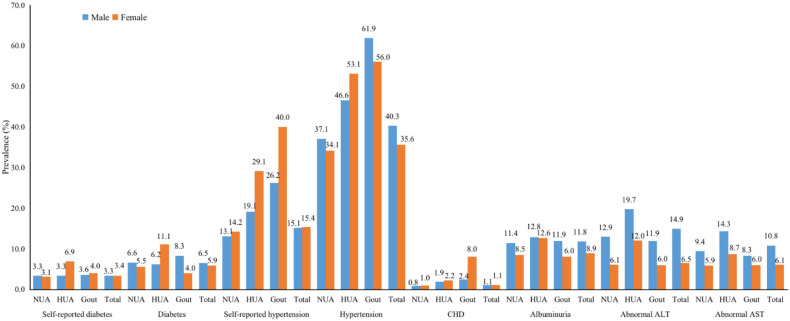
Sex-specific prevalence of common comorbidities in the study subjects separated by normouricemia, hyperuricemia, and gout groups. Abbreviations: NUA = normouricemia, HUA = hyperuricemia, CHD = coronary heart disease, ALT = alanine aminotransferase, AST = aspartate aminotransferase.

**Table 1 ijerph-19-16040-t001:** General characteristics and biochemistry test results of the study subjects specific for the normouricemia, hyperuricemia, and gout groups.

Variable	Normouricemia	Hyperuricemia	Gout	Total
Total	9144 (81.8)	1897 (17.0)	134 (1.2)	11,175 (100.0)
Age (years) *	55 (47–64)	54 (47–65)	56 (49–66)	55 (47–64)
Age group (years)				
35–44	1745 (19.1)	376 (19.8)	19 (14.2)	2140 (19.1)
45–54	2763 (30.2)	580 (30.6)	36 (26.9)	3379 (30.2)
55–64	2441 (26.7)	464 (24.5)	36 (26.9)	2941 (26.3)
65–74	2195 (24.0)	477 (25.1)	43 (32.0)	2715 (24.4)
Sex				
Male	3488 (38.1)	1493 (78.7)	84 (62.7)	5065 (45.3)
Female	5656 (61.9)	404 (21.3)	50 (37.3)	6110 (54.7)
Marital status				
Married	8092 (88.5)	1682 (88.7)	111 (82.8)	9885 (88.5)
Widowed	814 (8.9)	127 (6.7)	16 (11.9)	957 (8.5)
Single/divorced	238 (2.6)	88 (4.6)	7 (5.3)	333 (3.0)
Education level				
Primary school or lower	5271 (57.6)	794 (41.9)	69 (51.5)	6134 (54.9)
Middle school	2700 (29.5)	677 (35.7)	33 (24.6)	3410 (30.5)
High school	945 (10.3)	318 (16.8)	26 (19.4)	1289 (11.5)
College or above	228 (2.6)	108 (5.6)	6 (4.5)	342 (3.1)
Self-reported diabetes ^a^				
Non-diabetes	8854 (96.8)	1820 (95.9)	129 (96.3)	10803 (96.7)
Diabetics without OADS	75 (0.8)	22 (1.2)	1 (0.7)	98 (0.9)
Diabetics with OADS	215 (2.4)	55 (2.9)	4 (3.0)	274 (2.4)
Diabetes ^b^	541 (5.9)	138 (7.3)	9 (6.7)	688 (6.2)
Self-reported hypertension ^c^				
Non-hypertension	7884 (86.2)	1494 (78.8)	92 (68.7)	9470 (84.7)
Hypertension without medication	363 (4.0)	117 (6.2)	11 (8.2)	491 (4.4)
Hypertension with medication	897 (9.8)	286 (15.0)	31 (23.1)	1214 (10.9)
Hypertension ^d^	3224 (35.3)	912 (48.1)	80 (59.7)	4216 (37.7)
CHD				
Non-CHD	9060 (99.1)	1860 (98.0)	128 (95.5)	11048 (98.9)
CHD without medication	40 (0.4)	16 (0.8)	2 (1.5)	58 (0.5)
CHD with medication	44 (0.5)	21 (1.2)	4 (3.0)	69 (0.6)
Stroke	78 (0.9)	20 (1.1)	0 (0)	98 (0.9)
COPD	47 (0.5)	12 (0.6)	1 (0.7)	60 (0.5)
Cancer	24 (0.3)	5 (0.3)	0 (0)	29 (0.3)
Albuminuria	879 (9.6)	243 (12.8)	19 (14.2)	1141 (10.2)
Abnormal ALT	794 (8.7)	344 (18.1)	13 (9.7)	1151 (10.3)
Abnormal AST	659 (7.2)	250 (13.2)	10 (7.5)	919 (8.2)
SUA (μmol/L) *	307.0 (258.0–354.0)	470.0 (442.9–515.9)	436.0 (330.6–538.0)	326.0 (269.0–391.0)
Glu (mmol/L) *	4.6 (4.3–4.9)	4.7 (4.4–5.1)	4.6 (4.4–5.1)	4.6 (4.3–4.9)
SBP (mmHg) *	126.0 (114.5–140.0)	131.5 (121.0–145.5)	135.8 (123.1–148.5)	127.0 (115.5–141.0)
DBP (mmHg) *	80.0 (73.0–87.5)	84.0 (77.0–91.5)	86.0 (78.6–94.4)	80.0 (73.0–88.0)
BMI (kg/m^2^) *	23.0 (20.8–25.3)	24.7 (22.4–26.9)	24.6 (22.4–26.8)	23.3 (21.0–25.6)
WC (cm) *	81.0 (74.15–87.0)	87.0 (81.0–92.0)	86.0 (78.0–91.0)	82.0 (75.0–88.0)
TG (mmol/L) *	1.1 (0.8–1.6)	1.6 (1.1–2.4)	1.5 (1.1–2.2)	1.2 (0.8–1.8)
TC (mmol/L) *	5.0 (4.4–5.6)	5.1 (4.5–5.8)	5.3 (4.8–6.0)	5.0 (4.4–5.7)
LDL-C (mmol/L) *	2.9 (2.4–3.4)	2.9 (2.4–3.5)	3.1 (2.4–3.7)	2.9 (2.4–3.4)
HDL-C (mmol/L) *	1.4 (1.2–1.7)	1.2 (1.0–1.5)	1.3 (1.1–1.6)	1.4 (1.2–1.6)
ALT (μ/L) *	17.0 (13.0–23.0)	21.0 (15.0–29.0)	18.5 (14.0–25.7)	17.0 (13.5–24.0)
AST (μ/L) *	20.0 (17.0–24.0)	22.0 (18.0–27.0)	20.2 (17.0–24.0)	20.3 (17.0–25.0)

* Median (interquartile). ^a^ Self-reported previously diagnosed as hypertension by a doctor. ^b^ Self-reported previously diagnosed as hypertension by a doctor, or SBP ≥ 140 mmHg, or DBP ≥ 90 mmHg. ^c^ Self-reported previously diagnosed as diabetes by doctor. ^d^ Self-reported previously diagnosed as diabetes by doctor, or fasting blood glucose ≥ 7mmol/L. Abbreviation: OADS = oral antidiabetic drugs, CHD = coronary heart disease, COPD = chronic obstructive pulmonary disease, SUA = serum uric acid, Glu = glucose, SBP = systolic blood pressure, DBP = diastolic blood pressure, BMI = body mass index, WC = waist circumference, TG = triglyceride, TC = total cholesterol, LDL-C = low density lipoprotein, HDL-C = high density lipoprotein, ALT = alanine aminotransferase, AST = aspartate aminotransferase.

**Table 2 ijerph-19-16040-t002:** Prevalence of hyperuricemia by risk factors and major chronic disease, and the univariable and multivariable analyses results of the associated factors in males (gout patients were excluded, and normouricemia group is the reference).

Variables	Total (Prevalence, %) *	Normouricemia (Proportion, %)	Hyperuricemia (Proportion, %)	Crude OR (95% CI)	Adjusted OR (95% CI)
Total	4981 (30.0)	3488 (70.0)	1493 (30.0)	-	-
Age group (years)					
35–44	1106 (30.3)	771 (22.1)	335 (22.4)	1.00	-
45–54	1535 (32.1)	1043 (29.9)	492 (33.0)	1.09 (0.92–1.28)	-
55–64	1201 (27.6)	869 (24.9)	332 (22.2)	0.88 (0.73–1.05)	-
65–74	1139 (29.3)	805 (23.1)	334 (22.4)	0.95 (0.80–1.14)	-
Marital status					
Married	4505 (29.8)	3162 (90.7)	1343 (89.9)	1.00	-
Widowed	237 (30.8)	164 (4.7)	73 (4.9)	1.05 (0.79–1.39)	-
Single/divorced	239 (32.2)	162 (4.6)	77 (5.2)	1.12 (0.84–1.47)	-
Education level					
Primary school or lower	1835 (28.3)	1315 (37.7)	520 (34.8)	1.00	1.00
Middle school	1995 (29.7)	1403 (40.2)	592 (39.7)	1.07 (0.93–1.23)	1.03 (0.89–1.19)
High school	906 (31.2)	623 (17.9)	283 (19.0)	1.15 (0.97–1.37)	1.12 (0.94–1.35)
College or above	245 (40.0)	147 (4.2)	98 (6.5)	1.69 (1.28–2.22)	1.57 (1.18–2.1)
Diabetes					
No	4658 (30.1)	3258 (93.4)	1400 (93.8)	1.00	1.00
Yes	323 (28.8)	230 (6.6)	93 (6.2)	0.94 (0.73–1.21)	0.7 (0.54–0.91)
CHD					
No	4926 (29.7)	3461 (99.2)	1465 (98.1)	1.00	1.00
Yes	55 (50.9)	27 (0.8)	28 (1.9)	2.45 (1.44–4.17)	2.37 (1.36–4.13)
Hypertension					
No	2992 (26.6)	2195 (62.9)	797 (53.4)	1.00	1.00
Yes	1989 (35.0)	1293 (37.1)	696 (46.4)	1.48 (1.31–1.68)	1.20 (1.05–1.37)
Smoking in past 6 months					
No	2913 (31.2)	2005 (57.5)	908 (60.8)	1.00	1.00
Yes	2068 (28.3)	1483 (42.5)	585 (39.2)	0.87 (0.77–0.99)	0.83 (0.73–0.95)
Drinking in the past year					
No drinking	3695 (27.2)	2690 (77.1)	1005 (67.3)	1.00	1.00
Moderate drinking	229 (34.1)	151 (4.3)	78 (5.2)	1.38 (1.04–1.83)	1.62 (1.21–2.17)
Excessive drinking	1057 (38.8)	647 (18.6)	410 (27.5)	1.70 (1.47–1.96)	1.78 (1.53–2.08)
Drinking tea in the past year					
No	4060 (28.8)	2889 (82.8)	1171 (78.4)	1.00	-
Yes	921 (35.0)	599 (17.2)	322 (21.6)	1.33 (1.14–1.54)	-
BMI					
Normal	2584 (23.8)	1970 (56.5)	614 (41.1)	1.00	1.00
Underweight	284 (16.9)	236 (6.8)	48 (3.2)	0.65 (0.47–0.89)	0.75 (0.54–1.04)
Overweight	1653 (36.9)	1043 (29.9)	610 (40.9)	1.88 (1.64–2.15)	1.51 (1.30–1.76)
Obesity	460 (48.0)	239 (6.8)	221 (14.8)	2.97 (2.42–3.64)	1.84 (1.41–2.40)
Central obesity					
No	3859 (26.2)	2850 (81.7)	1009 (67.6)	1.00	1.00
Yes	1122 (43.1)	638 (18.3)	484 (32.4)	2.14 (1.87–2.46)	1.32 (1.10–1.58)
Visceral fat obesity					
No	2551 (23.6)	1950 (55.9)	601 (40.3)	1.00	-
Yes	2430 (36.7)	1538 (44.1)	892 (59.7)	1.88 (1.66–2.13)	-
Hyper-TG					
No	4138 (26.7)	3034 (87.0)	1104 (73.9)	1.00	1.00
Yes	843 (46.1)	454 (13.0)	389 (26.1)	2.35 (2.02–2.74)	1.57 (1.32–1.87)
Hyper-TC					
No	4437 (29.0)	3151 (90.3)	1286 (86.1)	1.00	1.00
Yes	544 (38.1)	337 (9.7)	207 (13.9)	1.51 (1.25–1.81)	1.24 (1.02–1.52)
Hyper-LDL-C					
No	4566 (29.7)	3210 (92.0)	1356 (90.8)	1.00	-
Yes	415 (33.0)	278 (8.0)	137 (9.2)	1.17 (0.94–1.44)	-
Hypo-HDL-C					
No	4105 (27.4)	2979 (85.4)	1126 (75.4)	1.00	1.00
Yes	876 (41.9)	509 (14.6)	367 (24.6)	1.91 (1.64–2.22)	1.53 (1.29–1.82)
Albuminuria					
No	4393 (29.6)	3091 (88.6)	1302 (87.2)	1.00	-
Yes	588 (32.5)	397 (11.4)	191 (12.8)	1.14 (0.95–1.37)	-
Abnormal ALT					
No	4237 (28.3)	3038 (87.1)	1199 (80.3)	1.00	-
Yes	744 (39.5)	450 (12.9)	294 (19.7)	1.66 (1.41-1.94)	-
Abnormal AST					
No	4440 (28.8)	3161 (90.6)	1279 (85.7)	1.00	1.00
Yes	541 (39.6)	327 (9.4)	214 (14.3)	1.62 (1.34–1.94)	1.48 (1.22–1.80)
SUA (μmol/L) ^▲^	371.0 (317.1–435.0)	340.0 (298.0–377.0)	474.8 (445.1–521.0)	-	
Glu (mmol/L) ^▲^	4.6 (4.3–5.0)	4.6 (4.3–4.9)	4.7 (4.4–5.1)	-	
SBP (mmHg) ^▲^	128.5 (118.0–141.5)	127.0 (117.0–140.5)	131.0 (121.0–144.0)	-	
DBP (mmHg) ^▲^	82.0 (75.5–90.0)	81.5 (75.0–89.0)	84.0 (77.0–92.0)	-	
BMI (kg/m^2^) ^▲^	23.3 (21.0–25.6)	22.8 (20.6–25.1)	24.5 (22.1–26.7)	-	
WC (cm) ^▲^	84.0 (77.0–90.0)	82.0 (76.0–88.0)	87.0 (81.0–92.0)	-	
TG (mmol/L) ^▲^	1.2 (0.9–1.9)	1.1 (0.8–1.7)	1.53 (1.0–2.3)	-	
TC (mmol/L) ^▲^	5.0 (4.4–5.6)	4.9 (4.3–5.5)	5.1 (4.5–5.8)	-	
LDL-C (mmol/L) ^▲^	2.9 (2.3–3.4)	2.9 (2.3–3.4)	2.9 (2.3–3.4)	-	
HDL-C (mmol/L) ^▲^	1.3 (1.1–1.5)	1.3 (1.1–1.6)	1.21 (1.0–1.5)	-	
ALT (μ/L) ^▲^	20.0 (15.0–27.3)	19.0 (15.0–26.0)	21.9 (16.0–30.0)	-	
AST (μ/L) ^▲^	21.3 (18.0–24.2)	21.0 (18.0–25.0)	22.0 (18.6–28.0)	-	

* The number in brackets is the prevalence of hyperuricemia in each specific group. ^▲^ Median (interquartile), only used category variables in the univariable and multivariable analyses. Abbreviations: CHD = coronary heart disease, SUA = serum uric acid, Glu = glucose, SBP = systolic blood pressure, DBP = diastolic blood pressure, BMI = body mass index, WC = waist circumference, TG = triglyceride, TC = total cholesterol, LDL-C = low density lipoprotein, HDL-C = high density lipoprotein, ALT = alanine aminotransferase, AST = aspartate aminotransferase.

**Table 3 ijerph-19-16040-t003:** Prevalence of hyperuricemia by risk factors and common comorbidities, and the univariable and multivariable analyses results of associated factors in females (gout patients were excluded, and normouricemia group is the reference).

Variables	Total (Prevalence, %) *	Normouricemia (Proportion, %)	Hyperuricemia (Proportion, %)	Crude OR (95% CI)	Adjusted OR (95% CI)
Total	6060	5656	404		
Age group (years)					
35–44	1015 (4.0)	974 (17.2)	41 (10.1)	1.00	1.00
45–54	1808 (4.9)	1720 (30.4)	88 (21.8)	1.22 (0.84–1.79)	1.01 (0.67-1.50)
55–64	1704 (7.8)	1572 (27.8)	132 (32.7)	1.99 (1.41-2.89)	1.48 (1.00–2.19)
65–74	1533 (9.3)	1390 (24.6)	143 (35.4)	2.44 (1.73–3.53)	2.26 (1.49–3.43)
Marital status					
Married	5269 (6.4)	4930 (87.2)	339 (83.9)	1.00	1.00
Widowed	704 (7.7)	650 (11.5)	54 (13.4)	1.21 (0.89–1.61)	0.95 (0.69–1.32)
Single/divorced	87 (12.6)	76 (1.3)	11 (2.7)	2.10 (1.05–3.83)	2.45 (1.24–4.84)
Education level					
Primary school or lower	4230 (6.5)	3956 (69.9)	274 (67.8)	1.00	1.00
Middle school	1382 (6.2)	1297 (22.9)	85 (21.0)	0.95 (0.73– 1.21)	1.28 (0.97–1.69)
High school	357 (9.8)	322 (5.7)	35 (8.7)	1.57 (1.07–2.24)	2.19 (1.47–3.29)
College or above	91 (11.0)	81 (1.5)	10 (2.5)	1.78 (0.86–3.32)	3.46 (1.66–7.24)
Diabetes					
No	5704 (6.4)	5345 (96.9)	359 (93.1)	1.00	-
Yes	356 (13.9)	311 (3.1)	45 (6.9)	2.15 (1.55–3.00)	-
CHD					
No	5994 (6.6)	5599 (99.0)	395 (97.8)	1.00	-
Yes	66 (13.6)	57 (1.0)	9 (2.2)	2.24 (1.10–4.55)	-
Hypertension					
No	3913 (4.8)	3725 (65.9)	188 (46.5)	1.00	1.00
Yes	2147 (10.1)	1931 (34.1)	216 (53.5)	2.22 (1.81–2.72)	1.42 (1.13–1.77)
Smoking in past 6 months					
No	6050 (6.7)	5646 (99.8)	404 (100.0)	- ^a^	- ^a^
Yes	10 (0)	10 (0.2)	0 (0)	- ^a^	- ^a^
Drinking in the past year					
No drinking	6024 (6.6)	5624 (99.4)	400 (99.0)	1.00	1.00
Moderate drinking	18 (22.2)	14 (0.2)	4 (1.0)	4.02 (1.32–12.26)	5.34 (1.66–17.18)
Excessive drinking	18 (0)	18 (0.4)	0 (0)	- ^a^	- ^a^
Drinking tea in the past year					
No	5566 (6.4)	5212 (92.1)	354 (87.6)	1.00	1.00
Yes	494 (10.1)	444 (7.9)	50 (12.4)	1.66 (1.20–2.24)	1.45 (1.04–2.01)
BMI					
Normal	3144 (3.4)	3038 (53.7)	106 (26.2)	1.00	1.00
Underweight	376 (2.7)	366 (6.5)	10 (2.5)	0.78 (0.38–1.44)	0.90 (0.46–1.74)
Overweight	1962 (9.6)	1773 (31.3)	189 (46.8)	3.06 (2.40–3.91)	2.42 (1.83–3.19)
Obesity	578 (17.1)	479 (8.5)	99 (24.5)	5.92 (4.43–7.92)	3.87 (2.69–5.57)
Central obesity					
No	4268 (4.3)	4086 (72.2)	182 (45.0)	1.00	1.00
Yes	1792 (12.4)	1570 (27.8)	222 (55.0)	3.17 (2.59–3.90)	1.30 (1.00–1.69)
Visceral fat obesity					
No	5614 (5.8)	5287 (93.5)	327 (80.9)	1.00	-
Yes	446 (17.3)	369 (6.5)	77 (19.1)	3.37 (2.56–4.40)	-
HyperTG					
No	5352 (5.2)	5075 (89.7)	277 (68.6)	1.00	1.00
Yes	708 (17.9)	581 (10.3)	127 (31.4)	4.00 (3.18–5.01)	2.16 (1.65–2.84)
HyperTC					
No	5302 (6.0)	4983 (88.1)	319 (79.0)	1.00	1.00
Yes	758 (11.2)	673 (11.9)	85 (21.0)	1.97 (1.52–2.53)	1.44 (1.09–1.89)
HyperLDL-C					
No	5491 (6.3)	5145 (91.0)	346 (85.6)	1.00	-
Yes	569 (10.2)	511 (9.0)	58 (14.4)	1.69 (1.25–2.24)	-
HypoHDL-C					
No	5606 (5.7)	5284 (93.4)	322 (79.7)	1.00	1.00
Yes	454 (18.1)	372 (6.6)	82 (20.3)	3.62 (2.76–4.69)	1.78 (1.3–2.45)
Albuminuria					
No	5526 (6.4)	5174 (91.5)	352 (87.1)	1.00	-
Yes	534 (9.7)	482 (8.5)	52 (12.9)	1.59 (1.16–2.13)	-
Abnormal ALT					
No	5666 (6.3)	5312 (93.9)	354 (87.6)	1.00	1.00
Yes	394 (12.7)	344 (6.1)	50 (12.4)	2.18 (1.57–2.96)	1.61 (1.15–2.26)
Abnormal AST					
No	5692 (6.5)	5324 (94.1)	368 (91.1)	1.00	-
Yes	368 (9.8)	332 (5.9)	36 (8.9)	1.57 (1.08–2.22)	-
SUA (μmol/L) ^▲^	290.0 (245.0–342.0)	284.0 (242.0–330.0)	457.0 (436.0–493.0)	-	
Glu (mmol/L) ^▲^	4.6 (4.4–4.9)	4.6 (4.3–4.9)	4.7 (4.5–5.0)	-	
SBP (mmHg) ^▲^	126.0 (113.5–140.5)	125.0 (113.0–140.0)	133.5 (120.5–147.6)	-	
DBP (mmHg) ^▲^	78.0 (72.5–86.5)	79.0 (72.0–86.5)	83.0 (75.9–90.0)	-	
BMI (kg/m^2^) ^▲^	23.3 (21.0–25.6)	23.1 (20.9–25.4)	25.4 (23.6–27.9)	-	
WC (cm) ^▲^	80.0 (74.0–87.0)	80.0 (74.0–86.0)	86.0 (81.0–92.0)	-	
TG (mmol/L) ^▲^	1.1 (0.8–1.7)	1.1 (0.8–1.6)	1.8 (1.2–2.6)	-	
TC (mmol/L) ^▲^	5.0 (4.4–5.7)	5.0 (4.4–5.6)	5.3 (4.6–6.0)	-	
LDL-C (mmol/L) ^▲^	2.9 (2.4–3.4)	2.9 (2.4–3.4)	3.0 (2.4–3.7)	-	
HDL-C (mmol/L) ^▲^	1.5 (1.2–1.7)	1.5 (1.2–1.7)	1.3 (1.1–1.5)	-	
ALT (μ/L) ^▲^	16.0 (13.0–21.0)	16.0 (12.6–20.3)	18.0 (14.0–26.0)	-	
AST (μ/L) ^▲^	20.0 (17.0–23.0)	20.0 (17.0–23.0)	21.0 (17.0–25.0)	-	

^a^ No female smoking and excessive alcohol consumption, those variables were excluded in univariable and multivariable analyses. * The number in brackets is prevalence of hyperuricemia in each specific group. ^▲^ Median (interquartile), only used category variables in the univariable and multivariable analyses. Abbreviations: CHD = coronary heart disease, SUA = serum uric acid, Glu = glucose, SBP = systolic blood pressure, DBP = diastolic blood pressure, BMI = body mass index, WC = waist circumference, TG = triglyceride, TC = total cholesterol, LDL-C = low density lipoprotein, HDL-C = high density lipoprotein, ALT = alanine aminotransferase, AST=aspartate aminotransferase.

## Data Availability

Not applicable.

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
