# Peer review of "High Prevalence of Hyperuricemia and Associated Factors among Zhuang Adults: A Cross-Sectional Study Based on the Ethnic Minority Population Cohort in the Southwestern China"

_ijerph, 2022, doi:10.3390/ijerph192316040_

Round 1

Reviewer 1 Report

Interesting results are adding information about hyperuricemia and comorbidities. Particularly interesting discrepancy between high prevalence of hyperuricemia and lower prevalence of gout.

Only remark can be related to figure 1. To increase informativity, only selected comorbidities can be included and items providing low information omitted.  

This article is devoted to the description of the prevalence of hyperuricemia and gout in a specific population characterised by ethnicity and location of living area. Prevalence data is obtained using cross-sectional region and ethnicity-specific general population survey and analysed considering measured well-known comorbidities. The result is compared with national sex and age adjusted prevalence data.

Strength. As hyperuricemia is related to genetic, lifestyle and other factors, descriptive data with associations to other health conditions and diseases are providing a good ground for propounding causal hypothesis which could be tested by analytic individual based longitudinal studies. Global variations in the occurrence of a disease or condition are usually a first step to advance to more detailed studies. Therefore, measuring such a variation is important for scientific knowledge. Reported data is obtained using sound epidemiologic methodology and conclusions are drawn appropriately taking into account typical limitations of such kinds of descriptive epidemiological studies. The level of analysis is appropriate for first-level description of associations (not going into deep adjustments).

Weaknesses. However, these authors mention this 'first-level analysis' as a limitation of the study. Also, the sampling approach looks more like convenience sampling and not a random sample (authors are not giving details on it), but this is often and typical approach for such kinds of studies due to practical obstacles. The authors mention generalizability as a limitation of the study. Also, possible variability of the measurements is mentioned. It would be desirable to have a more extensive discussion on results, e.g., just one sentence explaining interesting discrepancy between relatively high prevalence of hyperuricemia and low prevalence of gout: Are there any studies reporting the same?

3

·        Please, provide more information on sampling procedure, response rate, exclusions for analysis.

·        Please, add to the discussion about discrepancy mentioned above.

·        Figure 1.  No need to tell that this is a figure in title and explain how figure is built.

·        Figure 1.  Graphical presentation of all the results is not feasible. It could be that only selected results with the most visible differences are presented.

·        Possibly misprinting at line 271: 'health literacy' instead of ‘health literary’.

Reviewer 2 Report

The study presents the results of original research and presented results have not been published elsewhere. In section Materials and methods authors clearly describe the type of study and settings, participants, outcome measures and statistical analysis. In section Results authors clearly describe the sample which represents the remarkable size and representativeness of the sample. In section Discussion the authors logically link the results of this study to the results of previous studies. Conclusions are presented in an appropriate fashion and are fully supported by the data.

Author Response

Dear reviewer, we deeply appreciate your effort and patience in reviewing our manuscript, and thank you for your recognition of our research work.

Reviewer 3 Report

Dear Editor,

I carefully read the manuscript "High prevalence of hyperuricemia and associated factors among Zhuang adults: a cross-sectional study based on the ethnic minority population cohort in the southwestern China".

My comments and suggestions for the authors are the following:

 - English language needs to be carefully revised and improved.

 - The authors should specify the meaning of the abbreviations the first time they appear in the manuscript.

 - The authors refer to "hyper-TC". However, they should more properly refer to high levels of LDL-C. In effect, TC is not a recognized risk factor for the development of CVD (differently from increased levels of LDL-C).

 - The word "gender" should be replaced by "sex" throughout the manuscript. In effect, sex is usually categorized as female or male but there is variation in the biological attributes that comprise sex and how those attributes are expressed. Gender refers to the socially constructed roles, behaviours, expressions and identities of girls, women, boys, men, and gender diverse people.

 - Statistical analysis is well described and appropriately performed. However, the authors should specify if they considered as statistically significant either a 2-tailed or 1-tailed p-value <0.05 (line 150).

 - Line 153: A flow-chart detailing the process of selection of the population sample should be more properly included here.

 - In their manuscript, the authors should also refer to the findings from the Brisighella Heart Study group (i.e. doi: 10.1016/j.numecd.2020.03.005 , 10.1161/HYPERTENSIONAHA.119.13643 and doi: 10.1097/HJH.0000000000001927)

 - The limitations of the study should be further and more deeply discussed in the manuscript.

Round 2

Reviewer 3 Report

Dear Editor,

I carefully read the revised version of the manuscript, that is significantly improved in comparison with the original version. I warmly recommend its publication in the Journal.